# PAX6 Expression Patterns in the Adult Human Limbal Stem Cell Niche

**DOI:** 10.3390/cells12030400

**Published:** 2023-01-23

**Authors:** Naresh Polisetti, Günther Schlunck, Thomas Reinhard

**Affiliations:** Eye Center, Medical Center, Faculty of Medicine, University of Freiburg, Killianstrasse 5, 79106 Freiburg, Germany

**Keywords:** limbal stem cells, limbal niche cells, mesenchymal stem cells, melanocytes, limbal epithelial progenitor cells, limbal stem cell niche, PAX6

## Abstract

Paired box 6 (PAX6), a nuclear transcription factor, determines the fate of limbal epithelial progenitor cells (LEPC) and maintains epithelial cell identity. However, the expression of PAX6 in limbal niche cells, primarily mesenchymal stromal cells (LMSC), and melanocytes is scarce and not entirely clear. To distinctly assess the PAX6 expression in limbal niche cells, fresh and organ-cultured human corneoscleral tissues were stained immunohistochemically. Furthermore, the expression of PAX6 in cultured limbal cells was investigated. Immunostaining revealed the presence of PAX6-negative cells which were positive for vimentin and the melanocyte markers Melan-A and human melanoma black-45 in the basal layer of the limbal epithelium. PAX6 staining was not observed in the limbal stroma. Moreover, the expression of PAX6 was observed by Western blot in cultured LEPC but not in cultured LMSC or LM. These data indicate a restriction of PAX6 expression to limbal epithelial cells at the limbal stem cell niche. These observations warrant further studies for the presence of other PAX isoforms in the limbal stem cell niche.

## 1. Introduction

The limbal epithelium, bordering the corneal and the conjunctival epithelia, contains limbal epithelial stem/progenitor cells (LEPC), which are responsible for the homeostasis of the corneal epithelium, in its basal layers [1]. These LEPC reside in a specialized microenvironment, including blood vessels, nerves, and niche cells, composed primarily of mesenchymal stromal cells (LMSC) and melanocytes (LM), which protect adjacent LEPC from UV radiation [2,3]. The neural-crest-derived LMSC and LM exert various functions in limbal niche homeostasis by regulating LEPC maintenance, immune response, and angiostasis [4,5,6,7,8]. Any damage to LEPCs or disruption of the niche may lead to limbal stem cell deficiency (LSCD), a condition in which conjunctival epithelial cells overgrow the corneal surface, leading to its vascularization and opacification with subsequent loss of vision [9,10]. LSCD is seen in various diseases, including aniridia-associated keratopathy (AAK), which is caused by haploinsufficiency of the paired box 6 (PAX6) protein [11,12].

PAX6 is an evolutionarily conserved transcription factor that controls many downstream genes in eye morphogenesis and corneal homeostasis [13,14,15]. There are four PAX6 isoforms produced by alternative splicing [16]. The canonical PAX6-a protein includes three domains, two of which are DNA-binding domains, a paired domain (PD) and a homeodomain (HD), and a C-terminal proline, serine, and threonine (PST-rich) domain. The second isoform, PAX6-b, possesses an extra exon 5a in the PD domain, whereas the third isoform does not contain the PD domain (PAX6(ΔPD)). PAX6(S), the fourth isoform, contains the canonical PD and HD with a unique S tail at the C-terminus and is expressed only in the early stages of development [16]. PAX6 has been shown to determine the fate of LEPC and maintain corneal epithelial cell identity by controlling expression of cornea-specific keratins (KRT3/KRT12) [15,17]. The role of PAX6 in the limbal/corneal epithelium has been well documented; nevertheless, its expression in limbal niche cells has been mentioned in very few studies and is not entirely clear. Transient expression of PAX6 was noted in murine corneal stroma during development [18] and in adult bovine corneal stromal progenitors [19]. Recently, PAX6 expression was shown in isolated limbal niche cells of adult human corneal tissues, and it was found to alter neural crest progenitor cell status [20]. In contrast, Nishina and co-workers reported that there was no PAX6 expression in the human corneal stroma at 21 weeks of gestation [21]. Similarly, other groups reported on the absence of PAX6 expression in the adult limbal stroma [22] and in isolated limbal stromal cells [23]. PAX6 expression has been documented in human LM, and loss of PAX6 expression has been shown to influence melanogenesis in LM of AAK [12]. In contrast, Li and co-workers demonstrated that PAX6-negative cells in the limbal epithelial basal layer are positive for vimentin, suggesting either Langerhans cells or melanocytes [22]. To resolve this issue and acquire a better knowledge of the possible role of PAX6 in limbal niche cells, a thorough investigation of PAX6 expression in the limbal stem cell niche is necessary.

Thus, in the current study, we investigated PAX6 expression in fresh and organ-cultured human limbal tissues using immunostaining. The PAX6 protein expression was also examined in isolated LEPC, LMSC, and LM.

## 2. Materials and Methods

Human donor corneoscleral tissues (n = 3; mean age 75.2 ± 10.9 years; <16 h after death) not suitable for transplantation and organ-cultured corneoscleral tissue (mean age 60.3 ± 1.4, post-mortem duration 16.1 ± 6.1 h; culture duration 33.3 ± 1.4 d; n = 16 for cell isolation; n = 3 for immunohistochemistry) after retrieval of corneal endothelial transplants, with appropriate research consent provided by the Lions Cornea Bank Baden-Württemberg, were used as described previously [24]. The donor’s or their next of kin’s informed agreement to corneal tissue donation was obtained. Experiments using human tissue samples were approved by the Institutional Review Board of the Medical Faculty of the University of Freiburg (25/20) and adhered to the principles of the Helsinki Declaration.

### 2.1. Immunohistochemistry

Immunohistochemistry was carried out as previously described [25]. Briefly, corneoscleral tissue samples and organ-cultured corneoscleral tissue samples were frozen in liquid nitrogen after being embedded in an optimal cutting temperature (OCT) compound. Cryosections of 6–8 μm thickness were cut from the superior or inferior quadrants, fixed in 4% paraformaldehyde for 15 min, and blocked with 10% normal goat/donkey serum (NGS/NDS) in 0.3% Triton X−100 in PBS. Then, the samples were incubated in primary antibodies (Appendix A) diluted in 2% NGS/NDS, 0.1% Tween20, in PBS overnight at 4 °C or 2 h at room temperature. Alexa-488-,568-,558-conjugated secondary antibodies (Life Technologies, Carlsbad, CA, USA) were used to detect antibody binding, and the sections were mounted in Vectashield antifade mounting media with DAPI (Vector, Burlingame, CA, USA). A laser scanning confocal microscope (TCS SP−8, Leica, Wetzlar, Germany) was used to examine immunolabeled samples.

### 2.2. Cell Isolation and Culturing

LEPC (CD90^−^CD117^−^P-cadherin^+^), LMSC (CD90^+^CD117^+^P-cadherin^+^), and LM (CD90^−^CD117^+^P-cadherin^+^) from cadaveric limbal tissues were isolated and cultured as described previously [24]. Briefly, organ-cultured corneoscleral tissue was cut into 12 three-hour sectors, from which limbal segments were obtained via incisions 1 mm before and beyond the anatomical limbus. Limbal segments were enzymatically digested with collagenase A (Sigma-Aldrich, Roche; Mannheim, Germany; 2 mg/mL) for 18 h at 37 °C to produce cell clusters. Cell clusters were separated from single cells using 37 µm reversible cell strainers (Stem Cell Technologies, Köln, Germany). The cell clusters were then dissociated into single cells at 37 °C for 15–20 min with 0.25% Trypsin-EDTA (Gibco, Life Technologies, ThermoFisher Scientific, Karlsruhe, Germany). The single-cell suspensions were incubated for 5 min with FcR blocking reagent (Miltenyi Biotec, Bergisch Gladbach, Germany; 20 μL/10^6^ cells). Then, the cells were incubated for 40 min at 4 °C in the dark with mouse anti-human CD117-PE, CD90-APC, and P-cad-Alexafluor−488 antibodies (5 μL/106 cells) and their respective isotype controls in 100 µL phosphate-buffered saline (PBS), 0.1% sodium azide, and 2% fetal calf serum. FACS was performed using a FACS Aria II sorter (BD Biosciences, Heidelberg, Germany) and FACSDiva software (BD FACSDiva 8.0.1; BD Pharmingen; BD Biosciences). The post-acquisition data were analyzed using FlowJo software (FlowJo 10.2; Tree Star, Inc., Ashland, OR, USA). The sorted LM were seeded in T75 flasks (Corning, Tewksbury, MA, USA) coated in LN−511-E8 (iMatrix−511, Nippo; 0.5 µg/cm^2^) and cultured in CNT−40 medium (CellnTec, Bern, Switzerland). LEPC were seeded into T75 flasks in Keratinocyte serum-free medium supplemented with bovine pituitary extract and epidermal growth factor (Life Technologies, ThermoFisher Scientific, Karlsruhe, Germany). LMSC were seeded on a T75 flask in Mesencult media (Stem Cell Technologies). All the cultures were incubated at 37 °C, 5% CO_2_, and 95% humidity, with media replaced every other day.

### 2.3. Western Blotting

Western blotting was carried out as previously described [26]. Briefly, RIPA buffer (R0278, Sigma-Aldrich) with a protease inhibitor cocktail (complete Tablets Mini, Roche, Basel, Switzerland) was used to isolate total protein from cells. A colorimetric test (PierceTM BCA Protein Assay Kit, Thermo Fisher Scientific) was used to determine the total protein concentration. SDS-PAGE was used to separate 10 µg of total protein under reducing conditions, and immunoblot analyses were carried out with antibodies (Appendix A) against PAX6 (clone#poly901301—1:1000; clone#D3A9V—1:1000) and GAPDH (1:50,000), followed by horseradish peroxidase-labeled anti-mouse or rabbit IgG (Jackson ImmunoResearch Europe, Ely, UK). Enhanced chemiluminescence Western blot detection reagent (GE Healthcare, München, Germany) and FUSION FX imager/fusion software (Vilber Lourmat, Collégien, France) were used to visualize protein bands.

## 3. Results:

### 3.1. In Situ Localization of PAX6 in Non-Cultured Human Corneoscleral Tissues

To examine the PAX6 expression at the limbal stem cell niche, immunohistochemical staining of corneoscleral tissue (non-cultured) was performed with two distinct clones of PAX6 antibodies. Immunostaining revealed a nuclear localization of PAX6 (red) in the limbal epithelium (Figure 1A) and the corneal epithelium (dashed line delineates basement membrane (BM); Figure 1B). PAX6-negative cells (arrows, Figure 1A) were also observed in the basal layer of the limbal epithelium (arrows, Figure 1A) and occasionally in superficial layers of the cornea (arrowheads, Figure 1B). It is hypothesized that the loss of PAX6 expression in superficial epithelial cells associated with epithelial cell shredding necessitates the loss of cell–cell connections regulated by PAX6 [15]. PAX6 was not detected in the limbal or corneal stromal cells (Figure 1). The staining pattern for PAX6 was similar for both clones of the PAX6 antibodies (Figure 1).

To characterize PAX6-negative cells, we performed double immunostaining on corneoscleral tissue sections using epithelial (keratin marker, Pan-CK), stromal (vimentin), and melanocytic (Melan-A and human melanoma black−45 (HMB−45)) markers. Double immunostaining confirmed that PAX6-negative cells did not express epithelial keratins (Pan-CK, green) in the basal limbal epithelium (dashed line represents BM) but were positive for vimentin (arrow, green). PAX6 expression (red) was not observed in vimentin-positive limbal stromal cells (green, vimentin, Figure 2). Interestingly, PAX6-negative cells (arrows, Figure 2) were stained for the melanocyte markers Melan-A (green) and HMB-45 (green) at the basal layer of limbal epithelium. PAX6 expression appeared to be present in one of the Melan-A-positive cells (arrowhead, Figure 2), but this was due to a superimposed non-melanocytic cell, as revealed in the sequence of the confocal Z-stack images (see Appendix A for planes of Z-stack image, Appendix A). The PAX6-positive and -negative cells are listed in Table 1.

### 3.2. In Situ Localization of PAX6 in Organ-Cultured Human Corneoscleral Tissues

Immunohistochemistry was performed on organ-cultured corneoscleral tissue to examine the effect of culture conditions on PAX6 expression. Similar to fresh limbal tissues, PAX6-negative cells did not show keratin expression (green, Pan-CK) but expressed vimentin (green) and melanocyte markers Melan-A (arrows, green, Figure 3). A faint expression of vimentin was observed in basal limbal epithelial cells, which were positive for PAX6 (Figure 3). These data suggest that culture conditions induce vimentin expression in the basal cells of the limbal epithelium i.e., LEPC [27]. PAX6 expression was not observed in the limbal stroma (green, vimentin, Figure 3), similar to in fresh corneoscleral tissues.

### 3.3. PAX6 Expression in Cultured Limbal Cells

Isolated LEPC (P-cadherin^+^CD90^−^CD117^−^), LMSC (P-cadherin^+^CD90^−^CD117^−^), and LM (P-cadherin^+^CD90^−^CD117^−^) had similar cultural and functional characteristics as described previously (phase contrast micrographs, Figure 4A) [24]. The total protein from cultured LEPC (P1), LMSC (P1), and LM (P1) was subjected to Western blotting. In line with the immunohistochemical results, Western blotting demonstrated the presence of PAX6 (~50 Kda) solely in the population of LEPC (Figure 4B).

## 4. Discussion

The function of LEPC is known to be regulated by the limbal niche cells, LMSC, and LM, which maintain a balance between preservation and differentiation of LEPC [5,28,29]. Ocular surface alterations in AAK are caused by primary dysfunction and the gradual breakdown of the limbal stem cell niche due to PAX6-related effects on both melanogenesis and epithelial differentiation [12,30]. However, the expression pattern and function of PAX6 in limbal niche cells are controversial and ambiguous.

LMSC (also referred to as corneal stromal stem cells) were shown to support the regeneration of LEPC and to exhibit anti-inflammatory actions, selective migration to damaged tissue areas, and non-immunogenicity [5,7,29,31]. LMSC have also been used to treat ocular surface diseases such as dry eye, corneal burns, and limbal stem cell deficiency in experimental animal models [7,29,32,33]. Interestingly, a reduction of PAX6 in heterozygous (PAX6+/−) adult mice induced a severe defect in the corneal stroma and endothelium but had less impact on corneal epithelial cells with slightly delayed wound healing [34,35]. PAX6 expression was observed in the mouse corneal stroma during development [18], in adult bovine corneal stromal progenitors [19], and in human limbal niche cells [20]. These reports suggest that PAX6 may play an important role in limbal mesenchymal stem cell function. On the other hand, other groups reported that PAX6 expression was absent in the human corneal stroma at 21 weeks of gestation [21] and also in adult limbal/corneal stromal cells [22,23]. To resolve this conundrum, we used two different antibodies (different clones) raised against the C-terminal of PAX6, which is conserved in all PAX6 isoforms. We did not observe any signs of PAX6 expression in the limbal stroma (vimentin-positive stromal cells) of either fresh or organ-cultured corneal tissues. The isolated LMSC also did not show any expression of PAX6 with the different clones of antibodies used. Thus, this study demonstrates the absence of PAX6 expression in adult human limbal stromal cells. The discrepancies between the current study and earlier reports [20,36] could be attributed to differences in cell purity, i.e., epithelial cell contamination of cultures of LMSC [36], a paucity of investigations on in situ localization in fresh human corneoscleral tissues [20,36], and different imaging modalities (wide-field or confocal) used for detection. The small eye rat strain (bearing a PAX6 gene mutation) showed impaired migration of neural crest cells, although PAX6 transcripts were not observed in the neural crest derivatives by in situ hybridization [37]. This suggests that it is conceivable that PAX6 may not be expressed in mesenchymal stromal cells and may not directly regulate their migration and differentiation.

Melanocytes are neural-crest-derived cells, including LM, which are securely anchored to the BM, forming functional units with LEPC [6,8,24,26]. The primary role of LM is to produce melanin within specific organelles called melanosomes, which are then transmitted to neighboring epithelial stem/progenitor cells to protect them from UV light and free radicals, hence, minimizing oxidative-stress-induced cell damage [4,5,38]. Furthermore, recent research suggests that melanocytes and melanin pigments have immunosuppressive properties, modulating immunological responses by limiting lymphocyte proliferation and reducing the production of inflammatory cytokines [5]. Moreover, the dislocation of atypical melanocytes lacking melanosomes and melanin is linked to PAX6-related dysfunction of LM in AAK condition [12]. In contrast, the current study revealed that LM do not express PAX6, similar to the results of a previous study [22]. PAX6 inactivation in mice resulted in hypopigmentation of the retinal pigment epithelium due to direct effects on the expression of MITF (a master regulator of the pigmentation program) and other melanin-producing genes. However, the RPE and melanocytes produce melanin, but from separate embryonic origins, and their melanogenic properties differ [39]. PAX3, another paired box protein, stimulates and inhibits melanogenesis in neural-crest-derived epidermal melanocytes via the transcriptional regulation of MITF [40]. Moreover, a recent study reported that the PAX3 transcription factor is expressed by uveal and conjunctival melanocytes [41], which share the same anatomical site (ocular surface) as LM. This suggests that PAX3 may also govern melanogenesis in LM, which are derived from neural crest cells in a manner similar to epidermal and conjunctival melanocytes. However, the expression and role of PAX3 in LMs must be verified, which necessitates additional research.

In conclusion, PAX6 expression in the limbal stem cell niche is restricted to human limbal epithelial cells. This warrants further studies of other PAX isoforms in the limbal niche.

## Figures and Tables

**Figure 1 cells-12-00400-f001:**
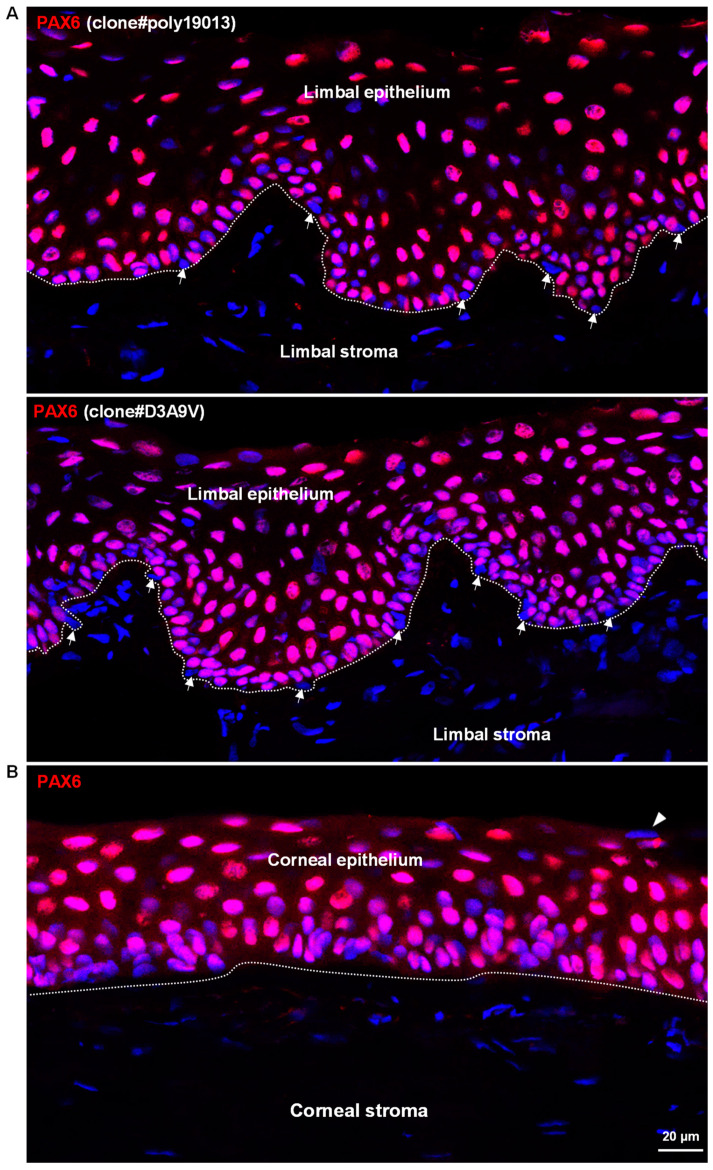
In situ localization of PAX6 -(**A**) Immunostaining of corneoscleral tissue sections showing the nuclear expression of PAX6 (red) in the limbal epithelium as well as PAX6-negative cells (arrows) in the basal layer of limbal epithelium and in the limbal stroma (dashed line represents basement membrane (BM)). Nuclear counterstaining with 4′,6-diamidino-2-phenylindole (DAPI, blue). (**B**) Immunostaining of the corneal tissue sections showing the PAX6 expression (red) in all epithelial cells of the cornea (except very few cells in the superficial layers, arrowhead) but not in the stroma (dashed line represents the basement membrane (BM)). Nuclear counterstaining with DAPI (blue).

**Figure 2 cells-12-00400-f002:**
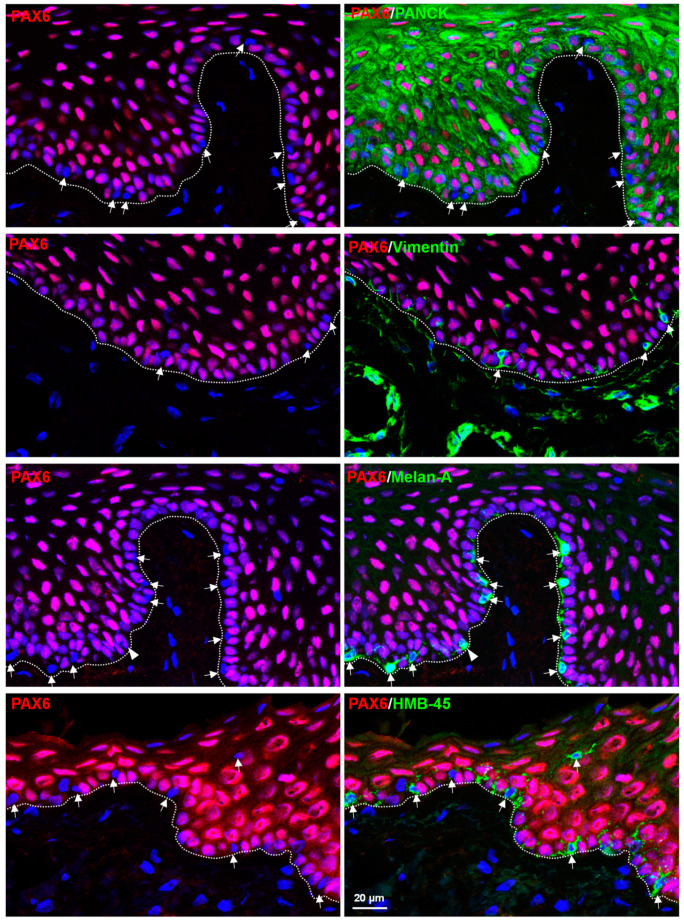
In situ co-localization of PAX6-Double immunostaining of corneoscleral sections showing PAX6-negative cells (arrows) not expressing epithelial keratins (Pan-CK, green) but expressing vimentin (arrow, green) and melanocyte markers Melan-A (green) and human melanoma black−45 (HMB−45, green) at the basal layer of limbal epithelium (dashed line represents basement membrane). Images of immunostaining also demonstrate that PAX6 was not expressed in the limbal stroma (vimentin-positive cells, green). PAX6 expression appeared to be present in one of the Melan-A positive cells (arrowhead, Figure 1), but this is due to a superimposed non-melanocytic cell, as revealed in the sequence of the confocal Z-stack images (see Appendix A for planes of Z-stack image). Nuclear counterstaining with DAPI (blue).

**Figure 3 cells-12-00400-f003:**
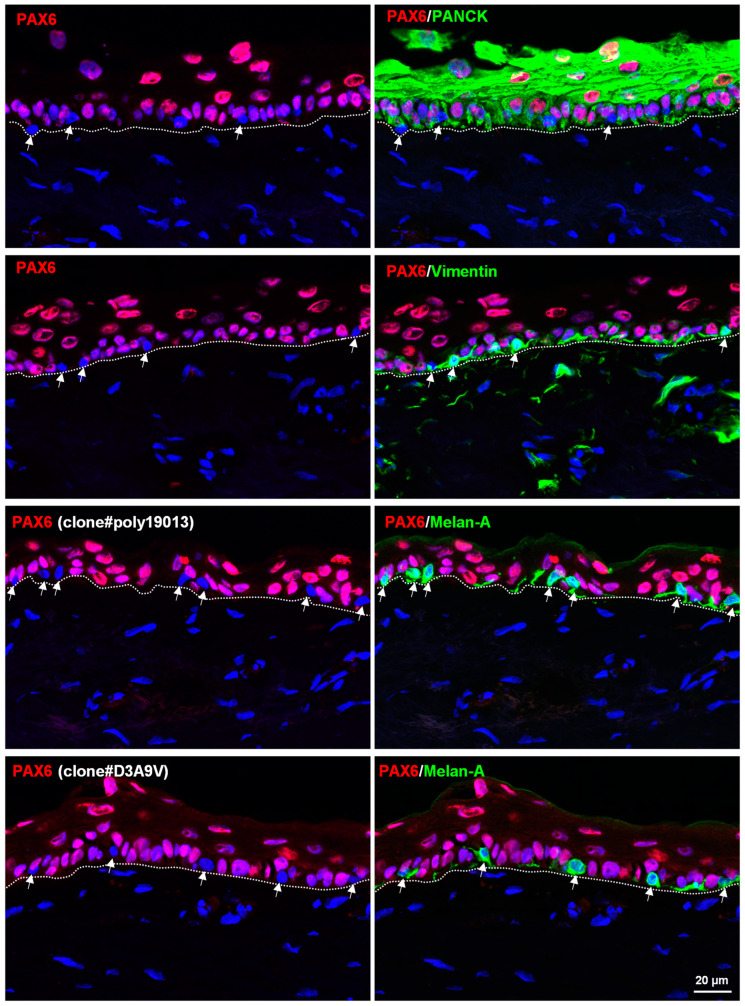
In situ localization of PAX6 in organ-cultured human corneoscleral tissues - Double immunostaining of organ-cultured corneoscleral sections showing the expression of vimentin (arrow, green) but not the epithelial keratins (Pan-CK, green) in PAX6-negative cells. PAX6 was not expressed by any of the cells in the limbal stroma (vimentin-positive cells, green). Immunostained limbal sections also showing the expression of melanocyte marker Melan-A (arrow, green) in the PAX6-negative cells at the basal layer of limbal epithelium. Nuclear counterstaining with 4′,6-diamidino-2-phenylindole (blue).

**Figure 4 cells-12-00400-f004:**
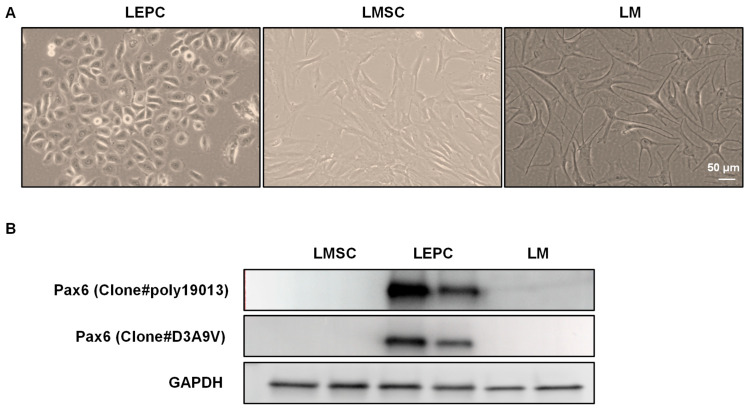
PAX6 expression on cultured limbal cells. (**A**) Phase contrast images showing the small cuboidal epithelial phenotype of limbal epithelial progenitor cells (LEPC, P1), spindle-shaped morphology elongated with prominent nucleolus of mesenchymal stromal cells (LMSC, P1), and large, flattened, smooth bodies with multiple dendrites of melanocytes (LM, P1). (**B**) Western blot analysis showing the PAX6 (both clones) expression in LEPC but not in LMSC or LM. Reprobing with an anti-GAPDH antibody served as a control. Uncropped versions of the Western blot are shown in Appendix A.

**Table 1 cells-12-00400-t001:** List of PAX6-positive and negative cells.

	Cornea				Limbus				
	Epithelial			Stromal	Epithelial			Stromal	Melanocytes
	Basal	Intermediate	Superficial		Basal	Intermediate	Superficial		
PAX6	+	+	+/−	−	+	+	+/−	−	−
Epithelial Keratins (PCK)	+	+	+	−	+	+	+	−	−
Vimentin	−	−	−	+	−	−	−	+	+
Melan-A	−	−	−	−	−	−	−	−	+
HMB45	−	−	−	−	−	−	−	−	+

−, undetectable; +, positivity.

## Data Availability

The datasets generated during and/or analyzed during the current study are available from the corresponding author on reasonable request.

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
