# Peer review of "PAX6 Expression Patterns in the Adult Human Limbal Stem Cell Niche"

_cells, 2023, doi:10.3390/cells12030400_

Round 1

Reviewer 1 Report

Manuscript number: cells-2164219

Title: PAX6 expression patterns in the adult human limbal stem cell niche

General comments

The manuscript is easy to read and the authors reveal important information concerning the PAX6 protein expression of different cell types in the adult human stem cell niche

1.       In the present study, the authors investigated the PAX6 protein expression using immunohistochemistry and western blotting

2.       The topic of the present research work is important because of the importance of PAX6 in cell differentiation and corneal homeostasis, particularly in some diseases like Aniridia-Associated Keratopathy (AAK).

3.       The work shows immunohistochemical stainings of cryosections and western blotting of cell lysates from limbal epithelial progenitor cells (LEPC), limbal mesenchymal stromal cells (LMSC) limbal melanocytes (LM). The description in the method section about cell isolation and culturing is very short. It could be helpful to explain in short the different media condition for the different cell types.

It is not entirely clear to me whether the number of cells isolated from 3 different donors is sufficient to perform FACS sorting, as described in reference 23, or whether the cultivation of melanocytes (LM = CD90-CD117+cadherin+) is possible without FACS sorting. A more detailed description would be desirable here.

4.       The conclusions drawn from the results of the study are conclusive and provide a basis for further experiments.

5.       The information in the reference list is appropriate.

6.       The illustrations are clear and well structured.

Author Response

We thank the reviewer for the kind appreciation of our work.

The work shows immunohistochemical stainings of cryosections and western blotting of cell lysates from limbal epithelial progenitor cells (LEPC), limbal mesenchymal stromal cells (LMSC) limbal melanocytes (LM). The description in the method section about cell isolation and culturing is very short. It could be helpful to explain in short the different media condition for the different cell types.

It is not entirely clear to me whether the number of cells isolated from 3 different donors is sufficient to perform FACS sorting, as described in reference 23, or whether the cultivation of melanocytes (LM = CD90-CD117+cadherin+) is possible without FACS sorting. A more detailed description would be desirable here.

Response: We sincerely apologize for the error. The number of donor samples (n=16) has been updated and the following detailed protocol has been added.

Briefly, organ-cultured corneoscleral tissue was cut into 12 three-clock-hour sectors, from which limbal segments were obtained by incisions made at 1 mm before and beyond the anatomical limbus. Limbal segments were enzymatically digested with collagenase A (Sigma-Aldrich, Roche; Mannheim, Germany; 2 mg/mL) at 37 °C for 18 h to generate clusters containing mixtures of epithelial, mesenchymal, and melanocytic cells. Cell clusters were separated from single cells by using reversible cell strainers with a pore size of 37 µm (Stem Cell Technologies, Köln, Germany). Subsequently, the cell clusters were dissociated into single cells with 0.25% Trypsin (Gibco, Life Technologies, ThermoFisher Scientific, Karlsruhe, Germany) containing 1 mM calcium chloride (PromoCell, Heidelberg, Germany) at 37°C for 15–20 min, and the obtained single cells from pooled corneoscleral tissues (4–6 cornea in single preparation) were used for further processing. The obtained single-cell suspensions were incubated with FcR blocking reagent (Miltenyi Biotec, Bergisch Gladbach, Germany; 20μL/106 cells) for 5 min. Following washing, cells were incubated with mouse anti-human CD117-PE, CD90-APC, and P-cad-Alexafluor-488 antibodies (5 μL/106 cells) and their respective isotype controls in 100µL phosphate-buffered saline (PBS), 0.1% sodium azide and 2% fetal calf serum for 40 min at 4°C in the dark. Following cell washing, FACS was performed using a FACS Aria II sorter (BD Biosciences, Heidelberg, Germany) and the FACSDiva software (BD FACSDiva 8.0.1; BD Pharmingen; BD Biosciences). FlowJo software (FlowJo 10.2; Tree Star, Inc., Ashland, OR) was used to analyze the post-acquisition data.

The CD90-CD117+P-cad+ sorted cells were seeded in LN-511-E8 (iMatrix-511, Nippo; 0.5 µg/cm2) coated T75 flasks (Corning, Tewksbury, MA,USA) and cultured in CNT-40 medium (CellnTec, Bern, Switzerland) at 37 °C, 5% CO2 and 95% humidity. CD90-CD117-/P-cad+ were seeded into T75 flasks in Keratinocyte serum-free medium (KSFM; 0.08 mM Ca2+) supplemented with bovine pituitary extract, epidermal growth factor (Life Technologies, ThermoFisher Scientific, Karlsruhe, Germany). CD90+CD117-/P-cad- were seeded on a T75 flask in Mesencult media (Stem Cell Technologies). All the cultures were incubated at 37°C, 5% CO2 and 95% humidity, and the media was changed every other day.

Reviewer 2 Report

The manuscript titled “PAX6 expression patterns in the adult human limbal stem cell niche” by

 Polisetti et al. are talking about the expression of Pax6, a transcription factor in the limbal area, using immunostaining and immunoblotting, in human donor corneoscleral tissues and culture cells. In this study, the authors concluded that the Pax6 expression is limited to the limbal epithelial cells and corneal epithelial cells. The manuscript is excellent in writing and organization. A few comments are as follows.

1.      What is the importance of using PANCK, vimentin, and Melan-A antibodies? It would be better to describe these proteins in the introduction of the manuscript.

2.      Based on the double immunostaining, it would be better to present in the tabular form where the Pax6 is expressed and where is not.

Reviewer 3 Report

The conclusion was made sense while a general knowledge. The data and the conclusion were not sufficient as a scientific manuscript.

I have some questions and sugestions about the manuscript.

1. As reported that there are at least 4 isoforms of the PAX6, that's PAX5a, PAX6, PAX6s and PAX6(Δ PD). In the introduction, the isoforms and structure of Pax6 should be presented.

2.The most intriguing is the expression and the function of the Pax6’s isoforms in the corneal epithelium, mesenchymal stromal cells and melanocytes. The PCR and the WB using the polyclonal Antibody of PAX6 could distinguish the isoforms, such as the Anti-PAX6 Antibody (ab265608)(Abcam). As presented in the top left figure of the Supplementary Figure 2 (Clone #poly 19013), the top left corner also could be distinguished.

3.There are some cells of co-expression Vimentin and PAX6, especially in organ-cultured human corneoscleral tissues. How explains this phenomenon?

Round 2

Reviewer 3 Report

Although the isoforms of the PAX6 should be future presented, the manuscprot is acceptable.